# The Magic of Proteases: From a Procoagulant and Anticoagulant Factor V to an Equitable Treatment of Its Inherited Deficiency

**DOI:** 10.3390/ijms24076243

**Published:** 2023-03-26

**Authors:** Juan A. De Pablo-Moreno, Andrea Miguel-Batuecas, María de Sancha, Antonio Liras

**Affiliations:** Department of Genetics, Physiology and Microbiology, School of Biological Sciences, Complutense University, 28040 Madrid, Spain

**Keywords:** coagulation, factor V, homeostasis, coagulopathies, proteases, rare diseases, equity

## Abstract

Proteostasis, i.e., the homeostasis of proteins, responsible for ensuring protein turnover, is regulated by proteases, which also participate in the etiopathogenesis of multiple conditions. The magic of proteases is such that, in blood coagulation, one same molecule, such as coagulation factor V, for example, can perform both a procoagulant and an anticoagulant function as a result of the activity of proteases. However, this magic has an insidious side to it, as it may also prevent the completion of the clinical value chain of factor V deficiency. This value chain encompasses the discovery of knowledge, the transfer of this knowledge, and its translation to clinical practice. In the case of rare and ultra-rare diseases like factor V deficiency, this value chain has not been completed as the knowledge acquisition phase has dragged out over time, holding up the transfer of knowledge to clinical practice. The reason for this is related to the small number of patients afflicted with these conditions. As a result, new indications must be found to make the therapies cost-effective. In the case of factor V, significant research efforts have been directed at developing a recombinant factor V capable of resisting the action of the proteases capable of inactivating this factor. This is where bioethics and health equity considerations come into the equation.

## 1. Proteases

Proteases, also known as peptidases, are enzymes that break proteins’ peptide bonds through hydrolysis [1]. They are widely distributed both in humans and in animals, in plants, bacteria, viruses, fungi, and parasites, and they are also used in the food and biopharmaceutical industries [2,3]. They are responsible for multiple tasks such as metabolic and functional regulation in general, biomolecule structuring and movement, catalysis, signaling, and transport through membranes [1,2,4,5,6,7,8,9].

### 1.1. Proteostasis

The homeostasis of cell proteins, known as proteostasis, allows precise control of protein synthesis, folding, conformational maintenance, and degradation to maintain appropriate turnover rates [10]. This process is coordinated with the different classes of chaperones and their regulators and ensures that cells can avail themselves of the proteins they need at the right time, minimizing the risk of misfolding or aggregation, which result in age-related proteinopathies such as neurodegenerative disorders like Alzheimer’s or Parkinson’s disease. The ability of cells to ensure proteostasis decreases with aging, which makes the body more susceptible to these conditions.

Maintaining balance within the proteome is not easy in the face of the accumulation of damage from exogenous and endogenous sources that occur during aging. This leads to a decreased ability to maintain proteostasis and to an impairment of the integrity of the proteome. The resulting accumulation of misfolded and aggregated proteins affects, above all, postmitotic cells such as neurons [11].

### 1.2. New Insights into Proteases, Biology, and Biomedicine

Proteases participate in untold physiological processes [12,13], regulating the activation, fate, location, activity, and turnover of multiple proteins [14]. Many stages of such fundamental processes as embryonic development, cell differentiation, proteomic degradation, inflammation, cellular senescence, apoptosis, cell homeostasis, and the coagulation cascade are regulated by proteases [12,15,16]. Consequently, an imbalance in the proteases’ proteolytic activity, i.e., an alteration in proteostasis, could result in serious pathological conditions such as cancer and neurodegenerative, inflammatory, and cardiovascular diseases [12,13,14,15].

At the present time, different protease-based strategies are under analysis, whose aim is to develop therapies for different conditions. [13,16]. Protease-based treatments are associated with high success rates given their high specificity, their ability to act at very low concentrations, and their low level of toxicity [2]. The interest in using proteases as pharmacological targets lie in the fact that, given that protease dysregulation is responsible for multiple serious conditions, restoring proper protease regulation could exert a therapeutic effect [13]. Collagenases (metalloproteinases) have been used to treat burns, repair cartilage, and address disc herniations, and matrix metalloproteins have been used to treat cardio-metabolic conditions with promising results [17].

A second approach is based on the use of protease inhibitors [13,14]. Angiotensin-converting enzyme inhibitors have been used to treat hypertension; HIV protease inhibitors as antiretroviral treatment; proteasome inhibitors to treat cancer; dipeptidyl peptidase IV inhibitors to treat type 2 diabetes; and thrombin and factor Xa inhibitors to activate blood coagulation [14,15,16].

## 2. Hemostasis and Its Regulation

The hemostasis system is made up of a series of perfectly coordinated cellular and biochemical mechanisms aimed at preventing blood loss following vascular damage through the formation of thrombi. These mechanisms also restore blood fluidity during the healing process [17,18]. The four components of hemostasis are the vascular endothelium, the platelet system, the coagulation cascade, and the fibrinolytic system.

Primary hemostasis [18,19,20] involves the formation of the white or platelet-rich thrombus. This process, mediated by platelets, the endothelium, and adhesion proteins, comprises activations, changes, and interactions between the different components of hemostasis. It also requires the participation of coagulation factor VII (FVII) [21,22,23]. Secondary hemostasis, also known as the coagulation cascade, is a process whereby soluble fibrinogen becomes soluble fibrin, which is stabilized and insolubilized by activated factor XIII (FXIIIa), known as the fibrin-stabilizing factor. This insoluble fibrin forms the stable red thrombus, made up of fibrin fibers and blood cells (leukocytes, platelets, and erythrocytes) [23,24,25,26,27].

Thrombin is the key molecule for hemostatic regulation as it interacts with multiple factors and components of blood plasma, in addition to activating FXI, factor XIII, factor VIII, and factor V. Other coagulation cascades inhibitory proteins such as activated protein C (APC), antithrombin (AT), and the tissue factor pathway inhibitor (TFPI) also participate in the homeostasis of the cascade [18,19].

Once bleeding has stopped and the endothelium has regained its integrity, fibrinolysis completes the hemostasis process restoring blood fluidity by clot lysis. Solubilization of the fibrin clot is essential for healing. Thrombin itself once again activates this process by prompting endothelial cells to stimulate the synthesis of the tissue plasminogen activator (t-PA) and the urokinase plasminogen activator (u-PA), which mediate the transformation of plasminogen into plasmin. Plasmin is ultimately responsible for the degradation of fibrin polymers. This plasmin formation process is also mediated by coagulation factors such as FXIa, FXIIa, and kallikrein. Fibrinolysis also has its own inhibitory regulators such as α2-antiplasmin, α2-antimacroglobulin, and the c1 esterase inhibitor [28]. 

The past few years have witnessed the generation of a significant body of knowledge about the physiological and molecular mechanisms of hemostasis, not only in adults but also in children. This body of knowledge has developed with particular intensity in the field of direct oral anticoagulants for children and adolescents, the prevention of pediatric venous thromboembolism, thromboprophylaxis, and the prediction of the risk of acquired thrombosis [28].

### The Homeostasis of Hemostasis

The concept of homeostasis [29,30,31] encompasses a variety of mechanisms that control the stability of the internal medium in terms of its composition and its physical and chemical variables. Hemostasis, on the other hand, refers to the mechanisms that ensure that the blood preserves its biological and physicochemical properties and that it keeps flowing through the circulatory system. The contrast of the two terms refers to the delicate balance between the coagulation system, in charge of neutralizing bleeding, and the fibrinolytic system, which prevents thromboembolism. For their part, both the coagulation cascade and fibrinolysis have their own homeostatic mechanisms and benefit from the assistance of proteins responsible for maintaining and ensuring the regulation of both processes.

Both the coagulation cascade and fibrinolysis involve a series of activating proteases whose function is regulated by other cofactors such as tissue factor (TF), thrombomodulin (TM), FVa, and FVIIIa. On the other hand, inhibitory proteases, such as APC, act as natural anticoagulants. This means that the homeostasis of the coagulation-fibrinolysis system is brought about by a balance of protease inhibitors and activators [32].

Moreover, a series of protease inhibitors such as serine protease inhibitors (SERPIN) [32], including AT, heparin cofactor II, protein-Z-dependent protease inhibitors (ZPI), protease nexin 1, and the C1 inhibitor exerts a negative control of coagulation and fibrinolysis. Other non-SERPIN anticoagulants, such as TFPI, the chief inhibitor of the TF-FVIIa complex, also play a role [33]. In addition, APC, together with protein S (PS), its main cofactor, are crucial physiological inhibitors of FVa and FVIIIa [34]. The complex formed by TM and thrombin activates protein C (PC), boosting the efficiency of coagulation when the endothelial PC receptor (EPCR) binds to PC’s Gla domain [35].

This sophisticated proteostasis system [36] is based on a “dispute” between activating and inhibitory proteases, which also occurs in the above-mentioned aging process, and the etiopathogenesis of neurodegenerative conditions. 

Recent cell-based theories, together with groundbreaking findings involving extracellular vesicles (EV), have sought to provide a holistic explanation of the intricate mechanisms inherent in in vivo coagulation. There is increasing evidence that cell- and platelet-derived EVs, which comprise microvesicles (MVs), exosomes, and apoptotic bodies, play an important role in the modulation of the coagulation cascade during hemostasis and thrombosis [37,38,39].

## 3. Coagulopathies: Dysregulation of Hemostasis

Disruption of the homeostasis of the hemostatic system may, in the medium- to long-term, result in a pathological situation [40]. Such disruptions may arise from a dysfunction in one or more coagulation factors (coagulopathies), spontaneous or inherited mutations, or some indirect cause such as an alteration in the components responsible for regulating the coagulation cascade, the fibrinolytic process, or endothelial function itself. These indirect causes may, in turn, be associated with mutations, with a decreased concentration in such cofactors as vitamin K [41], or with alterations at the sites where some molecules interact with the vascular endothelium.

The alteration of certain coagulation factors often results in an impairment in the function of that factor and, in the case of FV Leiden, in an exacerbated increase in coagulant function (hypercoagulability) [42]. Alterations may or may not lead to a dysregulation of coagulation factor activators or inhibitors, but the balance between the coagulation and fibrinolytic system is invariably disrupted. 

Many of the coagulopathies described, with the exception of von Willebrand’s disease, are what are known as rare diseases (RDs) or ultra-rare diseases (URDs). RDs have a prevalence of 1 in every 2000 individuals, while in the case of URDs, the prevalence is 1 in every 50,000 individuals [43,44]. Over 80% of RDs have a genetic origin [45]. According to the International Rare Disease Research Consortium (IRDiRC) and Orphanet, there are about 7000 RDs, only 60% of which have a known genetic background, encompassing about 4500 Mendelian phenotypes and 3500 genes in total. About half of all RDs are of autosomal origin, about 900 are X-linked, 60 are of mitochondrial origin, and 50 are Y-linked [43].

Ninety-five percent of RDs do not benefit from a specific and effective treatment, and for over 50% of these conditions, there is no specific diagnostic test that can be used as a basis for genetic counseling or prenatal or preimplantation diagnosis [43]. New next-generation mass sequencing and molecular cytogenetic techniques, particularly chromosomal microarray analysis, are showing considerable promise as effective tools to diagnose these conditions. 

Bleeding disorders may be associated with alterations in the coagulation cascade or platelet function, or they may be caused by a more general dysfunction such as a hepatic or autoimmune condition. Coagulopathies may be acquired or congenital. Congenital coagulopathies, brought about by an inherited alteration in the genes encoding different coagulation factors, may be autosomal or sex-linked, dominant or recessive, monogenic or polygenic [46,47,48]. Several coagulopathies have been described, including von Willebrand’s disease (von Willebrand factor deficiency), hemophilia A (FVIII deficiency), hemophilia B (FIX deficiency), FXI deficiency, and alterations in other coagulation factors such as fibrinogen, prothrombin, FV, FVII, and FX, among others [49].

### Factor V Deficiency

Congenital FV deficiency is a URD with an incidence of 1 to 9 per million live births [50,51]. It is transmitted in an autosomal recessive manner, with over 200 mutations of the FV gene (*F5*) having been described [52]. Most mutations have been found to occur in exon 13, the longest in the gene, which encodes the molecule’s B domain [53,54].

The clinical manifestations of FV deficiency usually present themselves at an early age and range from mild to severe. Symptoms include soft tissue or mucosal hemorrhages; fatal hemorrhages; profuse (nasal and menstrual) hemorrhages; bleeding during major and even minor surgical procedures and dental procedures; hematomas; and gastrointestinal, pulmonary, and intracranial hemorrhages. There is often no correspondence between FV plasma levels and the severity of symptoms [50,55,56]. Three degrees of factor V deficiency have been established: mild (>10% of normal plasma levels), moderate (1–10%), and severe (<1%) [50].

Diagnosis of the disease, which typically starts with an analysis of the patient’s medical history, is confirmed by means of coagulation assays and genetic tests. Coagulation assays usually measure prothrombin time (PT) and activated partial thromboplastin time (aPTT), as well as circulating FV levels. Molecular diagnosis is based on the detection of mutations specific to the affected family, given that FV deficiency is associated with a high consanguinity rate. Whole-genome sequencing is the method of choice to detect these mutations and determine whether they are pathological [45,50,57].

## 4. The New Pharmacology of Blood Coagulation: Alterations in Protease Activity

Newly developed medications for coagulation disorders often consist in customized treatments which, paradoxically, result in an alteration of protease-controlled homeostatic processes. 

Coagulopathies may result from a deficiency in a given coagulation factor but also from an alteration in the proteins that regulate coagulation, many of them proteases. In any event, whether the problem is a coagulation factor deficiency or an alteration in regulatory molecules, the end result is a disruption of homeostasis. The greater understanding available today of the different components of hemostasis makes it possible to develop strategies targeted at restoring normal homeostasis.

A congenital or acquired deficiency of the coagulant activity of a given clotting factor may be restored through exogenous administration of the deficient factor, with an extended half-life. A long half-life means that the molecule is resistant to its inactivating protease and therefore neutralizes its homeostatic action and keeps the levels of the deficient factor high. 

In addition, new hemostatic rebalancing therapies are currently underway at various clinical evaluation phases. The goal is to develop new pharmacological products aimed at restoring blood homeostasis by inhibiting some of the natural anticoagulant pathways, such as TFPI, AT-III, and APC. These agents will take the form of monoclonal antibodies, iRNAs, or protease inhibitors [58].

Subcutaneously administered anti-TFPI monoclonal antibodies are already showing promising results [59,60]. TFPI is an anticoagulant protein that may inhibit FXa directly and also indirectly by binding to FXa, forming a complex that interferes with the interaction between TF and FVIIa.

Other strategies are being evaluated for the treatment of hemophilia A and hemophilia B. One of them, which holds great promise, is based on the use of iRNAs to interfere with the synthesis of AT-III. AT-III is a SERPIN plasma glycoprotein responsible for regulating the proteolytic activity of several intrinsic and extrinsic pathway coagulation factors, specifically, FVII, IX, FX, XI, and thrombin [61,62].

Some protease inhibitors inactivate APC, rebalancing the coagulation cascade in patients with coagulopathies [63]. The function of APC, a natural anticoagulant produced and activated in the endothelium by thrombin and TM, is basically to inactivate FVIII and FV. Clinical trials have shown these protease inhibitors to be highly effective, particularly in patients with hemophilia A and FV deficiency, regardless of the severity of the disease. Given that APC is a natural regulator, it would seem sensible to use it for other bleeding disorders [64]. Patients with a congenital alteration of PC exhibit a dysregulation of the homeostasis between procoagulant and anticoagulant proteins, which predisposes them to thromboembolism [65], given the persistence of active procoagulant factors in the bloodstream. In these situations, the use of anticoagulants such as heparin, warfarin, aspirin, and clopidogrel should be the treatment of choice [65]. On the other hand, an autosomal dominant deficiency of PS, which is a cofactor of PC, is also, for the same reason, associated with the appearance of thrombotic events [66].

## 5. Factor V, a Procoagulant and Anticoagulant Protein

But how is it that proteases make it possible for a molecule like FV to act sometimes as a procoagulant and sometimes as an anticoagulant? This question can only be answered by taking a careful look at the molecule’s structure. It is a well-known fact that it is the structure of a molecule, especially if it is a protein, that determines and dictates its function. However, it must be taken into account that some proteins have an intrinsically disordered structure and yet perform their function without hindrance (intrinsically disordered proteins) [67,68]. In the case of blood coagulation, an initial signal, i.e., endothelial damage, is exponentially amplified by the presence of zymogens or precursor proteins that are irreversibly activated by the action of proteases [69,70]. The participation of several zymogens, which are sequentially activated in one same pathway, such as the coagulation cascade, boosts the efficiency of signal amplification [71].

The structure-function relationship that always governs molecular interactions is of great significance in this case. The structure of any coagulation factor must contain specific sites for binding to membranes and stabilizing proteins and for interacting with activating or inactivating proteases.

Endothelial cells generally play an indirect yet crucial role in the activation and inactivation of coagulation factors such as FVIII and FV. The homeostasis of the hemostatic process is precisely based on the maintenance of an appropriate balance between endothelial activity, the coagulation cascade, fibrinolysis, and the proteases responsible for modulating procoagulant processes.

It may thus be the case that, surprisingly, depending on the proteases at play and their plasma levels, the same coagulation factor, for e.g., FV, may play both a procoagulant and an anticoagulant role as a cofactor of the coagulation process [72,73,74,75].

In humans, the *F5* gene is located in region q23 of the long arm of chromosome 1 (1q23). It is a large gene of about 80 kb in size, which contains 25 exons and 24 introns, exon 13 being the largest. The gene’s mature messenger RNA, which encodes a protein made up of around 2224 amino acids, is approximately 6.8 kb in size [73,76,77,78]. In its inactive form, of approximately 330 kDa in weight [78,79,80], FV comprises six domains (Figure 1): A1, A2, B, A3, C1, and C2. The A1 and A2 domains constitute the heavy chain, and the B domain, entirely codified by exon 13, corresponds to the posttranslational region. The A3, C1, and C2 domains form the light chain of the molecule. Figure 1 shows thrombin-mediated FV activation with deletion of the B domain (Figure 1A) and its inactivation by APC and PS (Figure 1B).

The molecular structure of coagulation factors must provide binding sites for the factors or cofactors needed for the coagulation cascade to occur. Thus, the C1 and C2 domains of FVa are involved in the binding of FV to the platelet membrane [78,79], while the A1 and A2 domains of its heavy chain and the A3 domain of its light chain are responsible for the interaction of FVa with FXa to form the prothrombinase complex [79,81,82,83] (Figure 2A).

APC is responsible for coagulation homeostasis as it prevents hypercoagulability of the blood. Thrombin, which is the final product of coagulation, activates factors that precede it at different levels of the cascade and also reacts with APC through TM [84,85,86].

Inactivation of FVa by APC requires the participation of PS as a cofactor. FVa, APC, and PS come together at the membranes of either the endothelial cell or the platelet to form the APC-PS-FVa complex. Subsequently, APC binds to FVa’s Arg306, Arg506, and Arg679 residues bringing about the cleavage of the A2 domain, which results in the inactivation of FVa and the cessation of its procoagulant activity as a component of the prothrombinase complex [78,87,88].

The FV molecule is exceptional in that it is able to adapt its behavior depending on whether the body requires procoagulant or anticoagulant action. It carries out its anticoagulant function (Figure 2B) partly through the inactivation of the tenase complex and partly at the level of the initiation of the tissue factor pathway [74,75,87,89]. Inactivation of the tenase complex by FV is indirect as it is mediated by APC, which interacts with inactive FV and PS. This causes the inactivation of the FVIII component of the tenase complex by cleavage of its A2 domain. As for the anticoagulant effect of FV on the TF pathway, it is based on the formation of a ternary complex between TFPI, PS, and FV itself, which increases the anticoagulant capacity of TFPI [75,87].

Recently, low concentrations of a natural splicing variant called FV-short have been found to inhibit FXa, and hence the coagulation process, by interacting synergistically with TFPIa and protein S [90].

## 6. The Treatment of FV Deficiency: Bioethical Aspects of Health Equity

Treatment of FV deficiency is currently based on transfusions of fresh frozen plasma or the use of coagulation factor concentrates such as Octaplas^®^ [91,92], where factors are present in known amounts, facilitating optimal dosing. As in other coagulopathies such as hemophilia, antifibrinolytics such as tranexamic acid or β-aminocaproic acid are usually used, particularly in cases of mucosal hemorrhage [91].

In 2014, Drygalski et al. developed the first recombinant FV (^SUPER^FVa) [93], which constituted a major milestone and infused resewed hope in patients suffering from FV deficiency. ^SUPER^FVa is indeed capable of resisting APC-mediated inactivation as it was designed based on mutations at the level of the three APC cleavage sites (Arg306Gln, Arg506Gln, and Arg679Gln). This resistance to its inactivating protease increases ^SUPER^FVa’s specific activity three- or even fourfold, partly due to the introduction of a disulfide bond in its A2 and A3 domains. Its high levels of safety (low immunogenicity and absence of thrombogenic effects), efficacy and stability (long half-life), and optimal pharmacogenetic profile [94] make ^SUPER^FVa an ideal candidate for clinical trials. APC resistance is also an advantage in cases of acute traumatic coagulopathies, characterized by increased APC levels and hyperfibrinolysis [95]. However, this recombinant factor is still in the preclinical phase. 

From a pharmacologic point of view, one could wonder why a product as beneficial and safe as this one, which has obtained such good results in the preclinical phase, has not progressed to the clinical phase. To answer this question, the characteristics of rare and, particularly, ultra-rare diseases [96], such as FV deficiency, must be considered. An overwhelming proportion (95%) of these low-incidence diseases do not benefit from any specific or effective treatment options, and 50% of them even lack any diagnostic tests. Several countries have introduced legislation on RDs in order to facilitate a patient’s access to new treatments. At the same time, several platforms have been created (Orphanet, National Organization for Rare Disorders) to support patients and their families and promote the performance of studies and the development of treatments for these conditions. 

### 6.1. Bioethics and Health Equity

Bioethics regulates the four basic pillars of healthcare and biomedicine, namely beneficence (good medical practice), non-maleficence (not causing collateral damage), autonomy (freedom to choose and to act), and justice (equity in the distribution of resources). This last pillar is the most difficult to accomplish, both in the context of common conditions and, particularly, in the case of rare or ultra-rare ones. 

Achieving universal equity requires a paradigm shift and a commitment to diversity, equity, and inclusion (DEI). DEI has long been recognized as a crucial mechanism to extend the scope, creativity, and innovation capacity of research and healthcare with a view to resolving complex problems and reducing health- and disease-related inequalities [97]. Unfortunately, clinical trials are associated with a lack of diversity. Indeed, only 4–5% of subjects participating in drug-related clinical trials submitted for approval to the US Food and Drug Administration (FDA) between 1997 and 2014 belonged to historically underrepresented groups in medicine [98]. Clinical investigational protocols should be designed in such a way as to ensure that samples reflect the diversity of the populations affected by the disease under analysis.

A retrospective comparison of the progress achieved in the treatment of different conditions would show a surprisingly large difference not only between the development of treatments for common vs. uncommon conditions but also between a rare disease such as hemophilia A (factor VIII deficiency), which has an incidence of 1/6000 [99], and an ultra-rare disease such as FV deficiency, with an incidence of 1/1,000,000 [100] (Figure 3). In the 1930s, coagulation disorders, many of which had not yet been described or classified, were approached with palliative treatments based on the transfusion of whole blood, which was subsequently replaced by plasma. Between the 1960s and 1980s, the treatment of hemophilia was based, first, on low-purity plasma concentrates (cryoprecipitates), then on intermediate-purity concentrates, and finally, on high-purity concentrates. The 1990s and 2000s saw the advent of recombinant coagulation factors, which constitute the current palliative treatment of choice for hemophilia. In the case of FV deficiency, fresh frozen plasma, and plasma concentrates of several clotting factors (such as Octaplas^®^) remain the unspecific palliative treatment of choice.

### 6.2. From Costs to Equity

Investments in healthcare are necessarily finite. Funds tend to be made available based on the most urgent needs of the health system, which negatively impacts the ability to provide appropriate healthcare services. Taking into consideration the so-called “opportunity cost,” the introduction of a new treatment implies that if the budget available is not increased, some other healthcare interventions would have to be discontinued or, at best, adapted. The process followed to evaluate new healthcare technologies, including new medicines, new diagnostic techniques, or any other new procedure or device, involves gathering enough data to ensure that the implementation of such technologies is as fair and equitable as possible. The clinical, economic, ethical, and social aspects related to the new technology are typically analyzed, in an attempt to anticipate the short- and long-term consequences of its application, with efficiency considerations, usually governing the discussion. All of this process generally works very well for common diseases but is not so reliable for uncommon ones. 

As rare and ultra-rare diseases affect a small proportion of the population, little information is typically available to carry out an assessment of the relevant healthcare technologies. There is often scarce data on the progression of the disease, the adverse events derived from available treatments (if they exist), the number of patients affected, and, especially, the patient/year cost. This is particularly important as the cost of drugs for rare diseases is generally much higher than that of medicines for common conditions. The decisions required in these cases are often extremely difficult as, from an economic point of view, high prices stand in the way of beneficence. It is for these reasons that technology evaluation committees are increasingly interested in listening to the patients themselves (or their family members). 

### 6.3. The Clinical Value Chain of Factor V Deficiency

The clinical value chain extends from the discovery of new knowledge to its application to clinical practice and comprises several phases. The first one is the dissemination of the new knowledge to educational and research centers such as universities; the second one is the transfer of the new knowledge to companies through knowledge transfer support organizations; and the last phase is the clinical phase, which involves taking the new knowledge to the patient’s bedside. This is the sequence followed for the majority of technologies implemented in clinical practice. 

Nevertheless, things are different in the case of rare and ultra-rare diseases, as the value chain tends to break after the knowledge generation phase. This happens not because the advances made are not significant but rather because additional knowledge is often sought to ensure the cost-efficiency of the newly developed technology, i.e., the treatment of the disease.

This is particularly striking in the case of FV deficiency, where enough very promising breakthroughs have been made (Figure 4) to enable the commencement of the first phase of clinical trials. However, an overriding interest in looking for as many indications as possible, not just for patients with FV deficiency but also for patients suffering from bleeding episodes resulting from other causes, has prevented the findings made from progressing to further stages. 

Proteases are likely to play an important role in extending the range of applications of the already developed recombinant FV, which has been shown to be effective, safe, and pharmacokinetically stable. However, to what extent is the long wait that patients with FV deficiency must endure for a specific and effective treatment acceptable from a bioethical standpoint?

The solution to this dilemma might lie in the so-called *social pharmaceutical innovation*, as well as in the pursuit of alternative clinical research programs that move away from pre-established patterns and strive to channel more investments and clinical applications to rare diseases [101]. *Social pharmaceutical innovation* could take the form of drug repositioning (DR), which involves the identification of new indications for already approved medicines or the discovery of new therapies. This strategy may help reduce waiting times, costs, and the risks inherent in the discovery of new drugs, particularly for rare and ultra-rare diseases [102].

It would be essential to establish research networks for RDs comprising universities, research centers, and hospitals, so as to prioritize and promote cooperation and interactions between biomedical and clinical research groups, with particular attention to investigating the genetic, molecular, biochemical, and cellular underpinnings of RDs. This kind of research should lead to the development of new diagnostic and therapeutic tools to address low-prevalence diseases, in line with the goals of the International Rare Diseases Research Consortium, thus favoring the transfer of knowledge from bench to bedside [103]. This approach would undoubtedly help prioritize equity in the diagnosis and treatment of rare diseases.

## 7. Conclusions

Homeostasis of cell proteins, also known as proteostasis, allows precise control of protein turnover. Proteases, acting in coordination with one another and with certain chaperonins, are the protagonists of proteostasis. This process lessens with age, which may result in the development of conditions like Alzheimer’s, Parkinson’s, Huntington’s disease, cancer, and other dysfunctions.

In the field of hemostasis, which is the subject of this article, proteases activate one another in sequential order and as a function of other regulatory factors. The homeostasis of the coagulation-fibrinolysis system is based on a delicate balance between proteases and their activators and inhibitors. This sophisticated proteostasis system is based on a competition between activating and inhibitory proteases, which also occurs in other processes such as those involved in aging, or the etiopathogenesis of neurodegenerative diseases or cancer. 

Over just a few decades, proteases have ceased to be regarded as proteins exclusively responsible for regulating the turnover of other proteins and are now considered key players in the control of many physiological pathways. However, they are also the cause of innumerable diseases. This opens up opportunities for a wide range of molecular and pharmacological studies. 

The work done by proteases could be referred to as magical, given the meticulous way in which their functions are attuned to different physiological conditions. In the specific case of hemostasis, concurrent procoagulant and anticoagulant activities across multiple molecules or just within one single molecule, as in the case of FV, are regulated by proteases. At this point, the structure-function relationship acquires great importance as activation of a given coagulation factor can only occur if the structure of such factor contains specific cleavage sites for activating proteases. In addition, clotting factors must be equipped with other sequences capable of interacting with other molecules (many of them proteases) responsible for their inactivation.

All of the above evokes what could be called the insidious magic of proteases, whereby the only way of ensuring the economic viability of new therapies for rare or ultrarare diseases is to broaden their range of indications. Given that factor V deficiency is an ultra-rare disease, its clinical value chain has been protracted to the brink of breaking down. Indeed, the knowledge discovery phase regarding factor V and the treatment of factor V deficiency has stretched out over too many years, waiting for new applications and indications to be developed to increase the therapies’ cost-efficiency. This has consequently held up the transfer of the acquired knowledge to clinical practice. 

Although researchers came up with a recombinant factor V with optimal qualities for these patients many years ago, the product has not yet gone through the different phases of clinical evaluation as new indications must be found to make the therapy more cost-efficient. Proteases play an important role in this process as many of the new indications are based on making the new recombinant FV more resistant to APC so that it can be used as a treatment for other conditions such as acute traumatic coagulopathy, characterized by increased APC levels and hyperfibrinolysis.

This pharmaco-economic reality is, however, not compatible with the need for an effective treatment for patients suffering from FV deficiency, who are still awaiting the commencement of the first clinical trials of a recombinant FV that has been demonstrated to be highly effective for this condition. This is where bioethical, healthcare, and equity considerations come into play. 

Novel social pharmaceutical innovation strategies need to be implemented to help reduce the time, costs, and risks involved in discovering new drugs for rare and ultra-rare conditions.

## Figures and Tables

**Figure 1 ijms-24-06243-f001:**
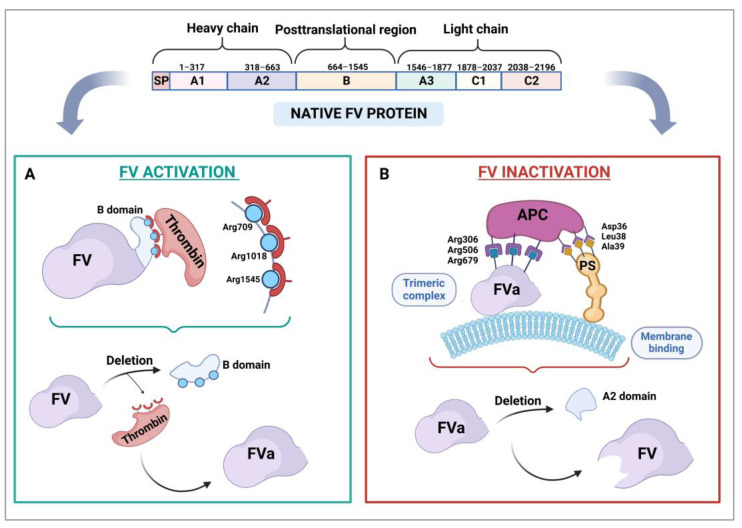
Activation and inactivation of coagulation factor V. The structure of the protein is made up of several domains: A1, A2, B, A3, C1, and C2. (**A**) Thrombin-mediated activation and deletion of the B domain. (**B**) Inactivation by APC and PS, assisted by vascular or platelet membrane phospholipids, and deletion of the A2 domain. FVa: activated factor V; APC, activated protein C; PS, protein S.

**Figure 2 ijms-24-06243-f002:**
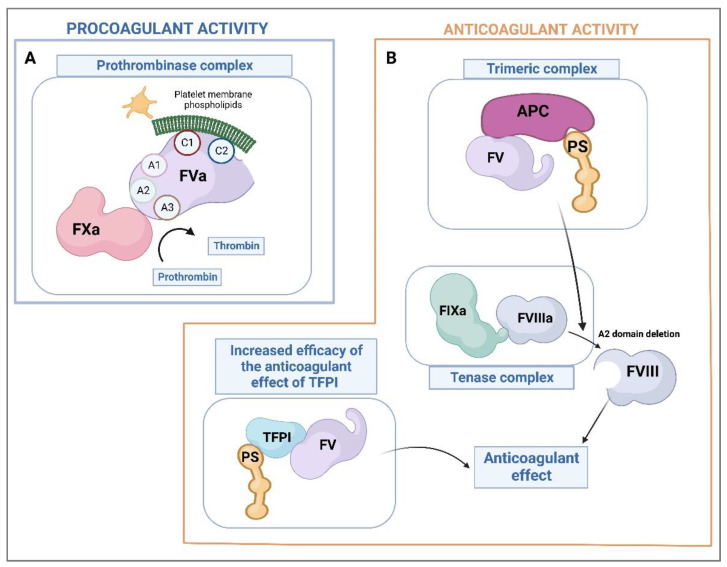
Factor V’s procoagulant and anticoagulant activity. (**A**) Prothrombinase complex-mediated procoagulant activity with thrombin generation. (**B**) Anticoagulant activity mediated by inactivation of the factor VIII component of the tenase complex and by the tissue factor pathway. F, factor; APC, activated protein C; PS, protein S; TFPI, tissue factor pathway inhibitor.

**Figure 3 ijms-24-06243-f003:**
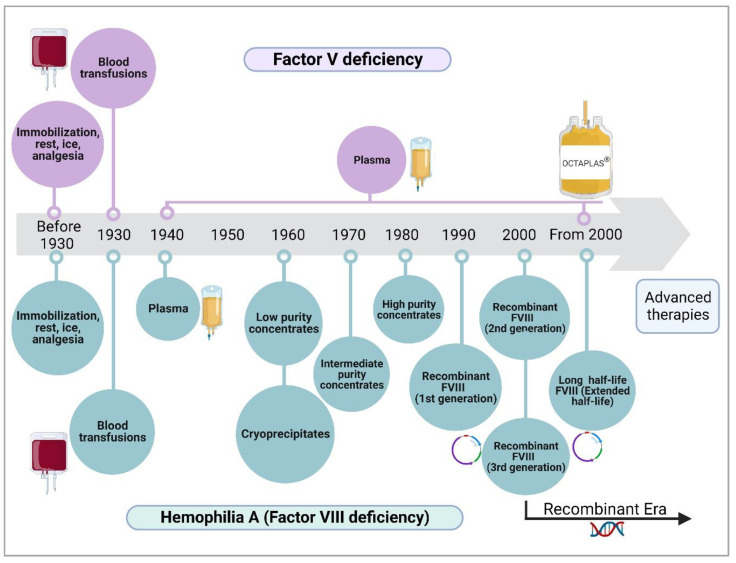
Evolution of the various strategies for the treatment of factor V deficiency and hemophilia A (Factor VIII deficiency).

**Figure 4 ijms-24-06243-f004:**
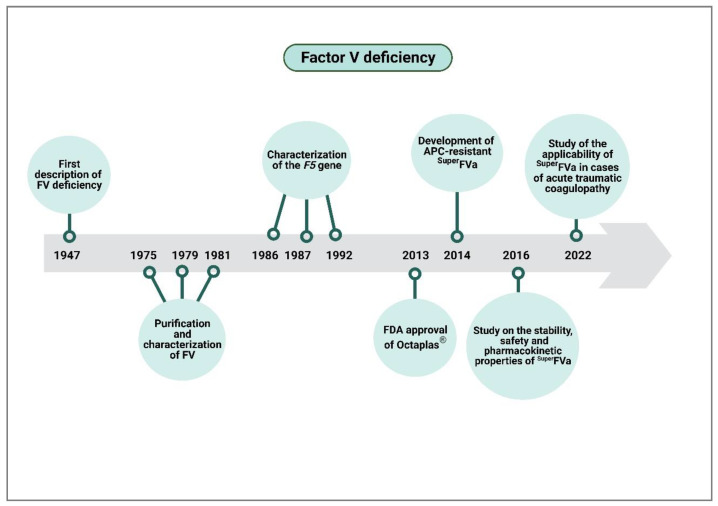
Clinical value chain of factor V deficiency. APC, activated protein C.

## Data Availability

Not applicable.

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
