# Peer review of "The Magic of Proteases: From a Procoagulant and Anticoagulant Factor V to an Equitable Treatment of Its Inherited Deficiency"

_ijms, 2023, doi:10.3390/ijms24076243_

Round 1

Reviewer 1 Report (Previous Reviewer 2)

The authors have followed the advice of the reviews. However, I still think the long introductory section on proteases distracts from the main message of the  paper, which is factor V. Lines 24-81 is all about proteases and reads like an undergraduate text book chapter. 

Criticism:

1, Figure 1 the upper part of this figure is not correct - the heavy chain does not include the B domain. 

2, the recent description of the splice variant FV-Short is missing. FV-Short has been shown to be an important synergistic TFPIa cofactor together with protein S. I suggest the authors consult the recent review on the topic:

Natural anticoagulant discovery, the gift that keeps on giving:finding FV-Short by Dahlback B, J Thromb Haemost, 2023 Feb 4:S1538-7836(23)00095-8. doi: 10.1016/jtha.s023.01.033. online ahead of print

Author Response

Response letter, 22 March 2023. Manuscript ID: ijms-2274494

Title: The magic of proteases. From a procoagulant and anticoagulant factor V to an equitable treatment of its inherited deficiency.

Journal: International Journal of Molecular Sciences

Special Issue: Peptidases: Role and Function in Health and Disease

Dear Editor,

I am enclosing a point-by-point response to reviewers’ comments. Changes to the text of the revised manuscript have been marked in red using Microsoft Word’s “track changes” function.

Point-by-point answers in this document are also marked in red.

The style, spelling and grammar has been revised, again, by a native speaker.

Reviewer #1 comments:

—The authors have followed the advice of the reviews. However, I still think the long introductory section on proteases distracts from the main message of the paper, which is factor V. Lines 24-81 is all about proteases and reads like an undergraduate textbook chapter.

Following the reviewer's indications, the introductory section on proteases has been further synthesized (Lines 31-45).

—Figure 1. The upper part of this figure is not correct - the heavy chain does not include the B domain.

In its inactive form, factor V has six domains (Figure 1): A1, A2, B, A3, C1 and C2. The A1 and A2 domains constitute the heavy chain; the B domain, encoded in its entirety by exon 13, constitutes the posttranslational region, and the A3, C1 and C2 domains form the light chain of factor V.

Figure 1 has been corrected along these lines.

—The recent description of the splice variant FV-Short is missing. FV-Short has been shown to be an important synergistic TFPIa cofactor together with protein S. I suggest the authors consult the recent review on the topic: Natural anticoagulant discovery, the gift that keeps on giving:finding FV-Short by Dahlback B, J Thromb Haemost, 2023 Feb 4:S1538-7836(23)00095-8.

Based on the reviewer's suggestion, the role of FV-short has been introduced and briefly discussed (Lines 322-324). A new reference (Dahlback B, now ref. #90) has been added accordingly (Line 732). Consequently, the numbering has been modified from this point onwards.

Reviewer #2 comments:

—Many papers that are cited in this work are review articles especially in the first part of the article. The authors should include more research articles. For example, lines 77-81, but also other places.

Based on the reviewer's suggestion, references have been updated throughout the manuscript and many review articles have been replaced by research articles.

—Figures included in the paper are supportive, however, in the text the authors should refer to appropriate panel (A, B etc.) not only to the Figure number.

Specific references to the figure panels have been added in text (Lines 283-284; 297 and 315)

—Figure 3, FV, plasma 1970 - do not understand why plasma is marked there. This kind of treatment was introduced in 1940. Please clarify.

We thank the reviewer for their observation. Figure 3 was indeed misleading in that regard. Figure 3 has been corrected to make it clear that plasma-based treatment started being administered in the 1940s and is still in use today.

Reviewer 2 Report (Previous Reviewer 3)

May papers that are cited in this work are review articles especially in the first part of the article. The authors should include  more research articles. For example lines 77-81, but also other places. 

 Figures included in the paper are supportive, however, in the text the authors should refer to appropriate panel (A, B etc.) not only to the Figure number.

Figure 3, FV, plasma 1970 - do not understand why plasma is marked there. This kind of treatment was introduced in 1940. Pleas clarify.

Author Response

Response letter, 22 March 2023. Manuscript ID: ijms-2274494

Title: The magic of proteases. From a procoagulant and anticoagulant factor V to an equitable treatment of its inherited deficiency.

Journal: International Journal of Molecular Sciences

Special Issue: Peptidases: Role and Function in Health and Disease

Dear Editor,

I am enclosing a point-by-point response to reviewers’ comments. Changes to the text of the revised manuscript have been marked in red using Microsoft Word’s “track changes” function.

Point-by-point answers in this document are also marked in red.

The style, spelling and grammar has been revised, again, by a native speaker.

Reviewer #1 comments:

—The authors have followed the advice of the reviews. However, I still think the long introductory section on proteases distracts from the main message of the paper, which is factor V. Lines 24-81 is all about proteases and reads like an undergraduate textbook chapter.

Following the reviewer's indications, the introductory section on proteases has been further synthesized (Lines 31-45).

—Figure 1. The upper part of this figure is not correct - the heavy chain does not include the B domain.

In its inactive form, factor V has six domains (Figure 1): A1, A2, B, A3, C1 and C2. The A1 and A2 domains constitute the heavy chain; the B domain, encoded in its entirety by exon 13, constitutes the posttranslational region, and the A3, C1 and C2 domains form the light chain of factor V.

Figure 1 has been corrected along these lines.

—The recent description of the splice variant FV-Short is missing. FV-Short has been shown to be an important synergistic TFPIa cofactor together with protein S. I suggest the authors consult the recent review on the topic: Natural anticoagulant discovery, the gift that keeps on giving:finding FV-Short by Dahlback B, J Thromb Haemost, 2023 Feb 4:S1538-7836(23)00095-8.

Based on the reviewer's suggestion, the role of FV-short has been introduced and briefly discussed (Lines 322-324). A new reference (Dahlback B, now ref. #90) has been added accordingly (Line 732). Consequently, the numbering has been modified from this point onwards.

Reviewer #2 comments:

—Many papers that are cited in this work are review articles especially in the first part of the article. The authors should include more research articles. For example, lines 77-81, but also other places.

Based on the reviewer's suggestion, references have been updated throughout the manuscript and many review articles have been replaced by research articles.

—Figures included in the paper are supportive, however, in the text the authors should refer to appropriate panel (A, B etc.) not only to the Figure number.

Specific references to the figure panels have been added in text (Lines 283-284; 297 and 315)

—Figure 3, FV, plasma 1970 - do not understand why plasma is marked there. This kind of treatment was introduced in 1940. Please clarify.

We thank the reviewer for their observation. Figure 3 was indeed misleading in that regard. Figure 3 has been corrected to make it clear that plasma-based treatment started being administered in the 1940s and is still in use today.

This manuscript is a resubmission of an earlier submission. The following is a list of the peer review reports and author responses from that submission.

Round 1

Reviewer 1 Report

Pablo-Moreno et al present a comprehensive review  on The magic of proteases. From a procoagulant and anticoagulant 2 factor V to an equitable treatment of its inherited deficiency. The review is valuable and need to benefit from proof reading by a fluent/native speaker of English..

Reviewer 2 Report

The review grasps over too many field and should be focused. What is the purpose of the paper? It starts by discussing proteases and reads more like a textbook on biochemistry. Then is discussed coagulation pathways in a superficial manner and then enters factor V, which is not a protease but rather a cofactor. The information on factor V is superficial and not up to date. I suggest the authors focus more on factor V and deletes the discussion on proteases.

Reviewer 3 Report

This is a very interesting, well-written paper, worth publication in IJMS. I just have a couple of suggestions, which should be considered by the authors.

At the end of the chapter 1. Author write about glutamy peptidase. Do they all have catalytic Glu136 residue?  Are they all inhibited by the discussed compound, or is the just an example? Please clarify and provide appropriate citation to this section.

Section 3.2 The last paragraph refers to recent theories and ground-breaking findings. Appropriate citations should be provided.

Section 5. Very often the function of the protein is correlated to its particular structure. However currently ii is well-known that many proteins perform their function being disordered. Please add short comment on that along with an appropriate citation